# CT-Detected Subsolid Nodules: A Predictor of Lung Cancer Development at Another Location?

**DOI:** 10.3390/cancers13112812

**Published:** 2021-06-04

**Authors:** Anton Schreuder, Mathias Prokop, Ernst T. Scholten, Onno M. Mets, Kaman Chung, Firdaus A. A. Mohamed Hoesein, Colin Jacobs, Cornelia M. Schaefer-Prokop

**Affiliations:** 1Department of Medical Imaging, Radboud University Medical Center, 6525 GA Nijmegen, The Netherlands; mathias.Prokop@radboudumc.nl (M.P.); ethscholten@gmail.com (E.T.S.); kaman.chung10@gmail.com (K.C.); colin.jacobs@radboudumc.nl (C.J.); cornelia.schaeferprokop@gmail.com (C.M.S.-P.); 2Department of Radiology and Nuclear Medicine, Academic Medical Center, 1105 AZ Amsterdam, The Netherlands; metsonno@gmail.com; 3Department of Radiology, Meander Medisch Centrum, 3813 TZ Amersfoort, The Netherlands; 4Department of Radiology, University Medical Center Utrecht, 3584 CX Utrecht, The Netherlands; f.a.a.mohamedhoesein@umcutrecht.nl

**Keywords:** computed tomography, lung neoplasm, biomarkers, epidemiology, risk

## Abstract

**Simple Summary:**

The risk assessment of pulmonary nodules on chest computed tomography (CT) is key to the early detection of lung cancer. Whereas a nodule’s morphological characteristics are strong predictors of malignancy for that individual nodule, it remains unclear whether the frequency and CT features of pulmonary nodules contribute to the prediction accuracy of future development of LC in another location. This was found by performing risk prediction modelling using age, sex, and CT measures from lung-cancer-free scans as predictors. We found that a greater number of part-solid and ground-glass nodules in the earliest scan was linearly associated with a higher risk of lung cancer developing in another location in the future. There were also indications that CT biomarkers of other pulmonary and heart diseases are risk factors for lung cancer. Our findings endorse the utilization of information from the entire scan to improve the accuracy of lung risk assessment.

**Abstract:**

The purpose of this case–cohort study was to investigate whether the frequency and computed tomography (CT) features of pulmonary nodules posed a risk for the future development of lung cancer (LC) at a different location. Patients scanned between 2004 and 2012 at two Dutch academic hospitals were cross-linked with the Dutch Cancer Registry. All patients who were diagnosed with LC by 2014 and a random selection of LC-free patients were considered. LC patients who were determined to be LC-free at the time of the scan and all LC-free patients with an adequate scan were included. The nodule count and types (solid, part-solid, ground-glass, and perifissural) were recorded per scan. Age, sex, and other CT measures were included to control for confounding factors. The cohort included 163 LC patients and 1178 LC-free patients. Cox regression revealed that the number of ground-glass nodules and part-solid nodules present were positively correlated to future LC risk. The area under the receiver operating curve of parsimonious models with and without nodule type information were 0.827 and 0.802, respectively. The presence of subsolid nodules in a clinical setting may be a risk factor for future LC development in another pulmonary location in a dose-dependent manner. Replication of the results in screening cohorts is required for maximum utility of these findings.

## 1. Introduction

Lung cancer (LC) is the top cause of cancer deaths worldwide, responsible for 2.1 million deaths in 2018 [1]. A number of 57% of the patients are diagnosed with metastatic disease, for which the 5-year survival rate is approximately 5% [2]. The early diagnosis of lung cancer almost always involves the early detection and risk assessment of pulmonary nodules on chest computed tomography (CT).

Based on their CT attenuation, lung nodules are categorized as either solid or subsolid (SSN). The great majority of lung nodules are solid with a density in the range of other soft tissue structures, such as muscles. They are very common in a screening population of people that have been smoking for a number of years [3,4,5,6,7]. SSNs are divided into pure ground-glass and part-solid nodules: ground-glass nodules have a vaguely increased CT attenuation, which preserves the visibility of bronchial and vascular margins; part-solid nodules have a ground-glass as well as a solid component.

Prior studies have shown that SSNs are less prevalent but at greater risk of being or becoming malignant compared to solid nodules [8,9,10]. To date, one study in a LC screening setting reported that the presence of SSNs might be a risk factor for LC development in another location in the lungs [4]. The main goal of our study was to attempt to reproduce these findings in a clinical setting, which may consist of any members from the general population rather than a high-risk population eligible for screening. Namely, we investigated whether the presence and frequency of certain nodule types in a baseline CT scan can predict the development of a LC that has not yet (visibly) originated, and what the added value of this information is on top of other LC risk predictors.

## 2. Materials and Methods

### 2.1. Study Population

Anonymized patient information and CT scans from between January 2004 and December 2012 were collected from all patients 40 years or older from the University Medical Center Utrecht (Center A) and Radboud University Medical Center (Center B) in the Netherlands. These data were cross-referenced with the Dutch Cancer Registry to determine the patients who were diagnosed with LC up to December 2014. Informed consent was waived by the institutional review board.

All CT scans from patients diagnosed with LC were retrieved, and the earliest available CT scan from a sample of approximately three times as many LC-free patients was selected at random. Patients and/or scans were subsequently excluded if there were image-retrieval errors, if the image was considered to be of too low quality by a radiologist, if the scan-slice thickness was greater than 3 mm, or if no scan was available at least 2 months prior to the LC diagnosis; the latter criteria was selected arbitrarily to be able to ensure the correct identification of the malignant nodule without excluding too many patients.

### 2.2. CT Features

All collected scans from Centers A and B were assessed independently by radiologists O.M.M (9 years’ experience) and E.T.S. (>20 years’ experience), respectively. Their first task was to extract CT features as given in Table 1, which can be divided into two groups: nodule-type variables and control variables, the latter of which was used to adjust for potential confounding. Dedicated in-house software (version 19.9 of CIRRUS Lung Screening, DIAG, Radboudumc, Nijmegen, The Netherlands) was applied to the scans to automatically detect and volumetrically segment pulmonary nodules. The radiologists had to either accept or reject non-calcified nodule candidates provided by the software, and were additionally required to manually scroll through the full scan for potential nodules which had been missed by the software. Only nodules that would trigger a follow-up scan according to nodule management recommendations [11,12,13] (BTS, LUNG RADS, Fleischner) and perifissural nodules (PFN) were included in the analysis [14]; nodules related to an obvious infection (multifocal, unclearly defined, or confluent SSNs) were excluded [15].

Each nodule was given a unique ID; this enabled the radiologists to retrospectively assess a nodule over time in patients who had been diagnosed with LC. This was relevant for the second task of identifying the malignant nodule. This was done by assessing nodule morphology, growth pattern, and location; the only relevant information provided by the Dutch Cancer Registry was the time of LC diagnosis and the lobe location. A confidence score from 1 to 4 was used to determine the nodule most likely to be the LC: 1 indicated benign morphology, 2 indicated uncertainty due to multiple nodules present, 3 indicated suspicious morphology without a CT scan within 1 month of LC diagnosis, and 4 indicated suspicious morphology with a CT scan within 1 month of LC diagnosis. Only nodules with a score of 3 or 4 were included as cancer cases in this study.

Nodules were classified into one of four mutually exclusive subtypes: solid, part-solid, ground-glass, or PFN. Part-solid and ground-glass nodules fall into the group of SSNs, and the definition of a perifissural nodule—which show morphology indicative of benign intrapulmonary lymph nodes—was that as defined by Hoop et al. [16].

Only the CT features of each patient’s earliest scan were considered for the risk prediction analysis. The derivation cohort included all eligible LC-free patients and the eligible LC patients in which it could be said with high confidence that no malignant nodule was retrospectively visible in the earliest scan. Some LC-free patients were diagnosed with other cancer types which may have metastasized to the lungs. LC-free patients who were highly suspect of having pulmonary metastatic disease were excluded.

### 2.3. Statistical Analysis

All statistical analysis was performed in the statistical program R version 3.6.3 [17]. Weighted Cox proportional hazards regression was performed [18], with LC diagnosis as the event and the time between the earliest scan to LC diagnosis or the end of follow-up as the time to event. Weights were determined based on Barlow’s estimate computing the Cox model for case–cohort designs [19]. Since all patients diagnosed with LC were considered, they were given a weight of 1. LC-free cases were given a weight of 37,085/1779 = 20.8, i.e., the total number of LC-free cases in the full cohort divided by the number selected at random. A Kaplan–Meier curve was plotted to visualize the time-dependent relationship between nodule types present and LC incidence risk. The Kaplan–Meier curve was adjusted using rescaled inverse probability weighting based on the following untransformed variables [18]: age, sex, emphysema, bronchitis, interstitial lung disease, lymphadenopathy, aortic calcifications, and coronary calcifications.

Univariable analysis of all independent variables was performed first (Table 1). Transformations of continuous variables with first- or second-degree factorial polynomials were considered [20,21]; transformations were reverted if there were indications for model overfitting during interval validation. Multivariable regression of all variables (i.e., the Full Model) is visualized in the form of a forest plot to indicate which predictors appear to be strongest. To gauge the best predictive performance achievable without nodule-type variables, a parsimonious multivariable model—in which variables which do not contribute significantly to the model are excluded (*p* > 0.20)—containing only control variables was derived (i.e., the Control Model). The backwards elimination procedure was used for variable selection. Interaction terms between the largest nodule’s type and its mean diameter were considered, but no interaction was found. Finally, a second parsimonious model including nodule-type variables was derived (i.e., the Nodule Type Model) to assess the added value of nodule-type information compared to the Control Model.

Results are presented in terms of hazard ratios with 95% confidence intervals. The Schoenfeld global test was used to test the proportional hazards assumption [22]. Internal validation was performed for each model by deriving new model coefficients on 1000 weighted bootstrap samples and calculating optimism as the difference between the bootstrap model performance on the bootstrap sample and on the original cohort. With the exception of determining variable inclusion in a parsimonious model, statistical significance was defined as a *p* value of less than 0.05.

## 3. Results

We considered 186 LC patients and 711 LC-free patients from Center A and 292 LC patients and 508 LC-free patients from Center B with adequate CT scans (Figure 1). The malignant nodule was already visible in the earliest available scan in 315 of the 478 LC patients, and 41 of the 1219 LC-patients were suspected of having pulmonary metastasis. In total, 163 (34%) LC patients and 1178 (97%) LC-free patients were included in the derivation cohort. The median time to end of follow-up or LC diagnosis was 1923 days (twenty-fifth and seventy-fifth percentiles = 1148 and 2828 days). The patient and CT demographics are summarized in Table 1.

### 3.1. Univariable and Full Multivariable Analysis

The continuous variables, age and solid nodule count, were transformed into second-order polynomials for a better fit of their non-linear relationship with future LC risk. A univariable analysis found all independent variables to be statistically significant predictors of future LC development, with three exceptions: sex, presence of a perifissural nodule, and perifissural nodule count (Appendix A).

In the Full Model, the presence of at least one solid nodule was a future LC risk factor, but the risk decreased as the number of solid nodules increased (Figure 2). Additionally, the number of part-solid and ground-glass nodules was linearly correlated to the risk. The presence of at least one perifissural nodule was a protective factor, but was not statistically significant in the Full Model (*p* = 0.07). Compared to univariable analysis, four other variables were no longer considered to be significant predictors, namely “nodule diameter of the largest nodule”, “largest nodule is a SSN”, “presence of a part-solid nodule”, and “presence of a ground glass nodule” (Figure 2).

### 3.2. Parsimonious Lung Cancer Incidence Risk Models

Only “nodule diameter of the largest nodule” was excluded from the parsimonious Control Model, leaving in age, sex, emphysema, bronchitis, interstitial lung disease, lymphadenopathy, aortic calcifications, and coronary calcifications (Appendix A). Four nodule-type variables were added to form the Nodule Type Model (Table 2): “solid nodule count” (transformed), “part-solid nodule count”, “ground glass nodule count”, and “presence of a perifissural nodule”, the latter indicating a lower future LC risk.

Given the finding that there may be a dose-dependent relationship between the number of SSNs present in a scan, each cohort was divided into four groups: patients with no nodules, patients with at least one nodule (none of which are SSNs), patients with a single SSN, and patients with more than one SSN. The LC-free survival probability over time of the groups was visualized as Kaplan–Meier curves (Figure 3). Among patients with at least 5 years of LC-free follow-up or LC diagnosis within 5 years, LC incidence was 6% (23/379) in those without any nodules at baseline, increasing to 22% (67/300) in those with at least one solid nodule (no subsolid nodules), 22% (18/82) in those with a single SSN, and 40% (14/35) in those with at least two SSNs. The difference in LC-free survival between the group without nodules and the other three groups was statistically significant (log-rank test *p* < 0.001). The difference between having at least two SSNs compared to a single SSN or only solid nodules was also significant (*p* = 0.05 and *p* = 0.03, respectively). The LC probability between the single SSN and only solid nodules groups was equivalent (*p* = 1.00). Weighted 5-year LC-free survival probabilities are given in the Figure 3 legend.

Internal validation of the Nodule Type Model (Table 2) resulted in an area under the receiver operating characteristic curve (AUC) of 0.827 (95% confidence band = 0.826 to 0.827) on the original sample and 0.838 (0.837 to 0.838) on the bootstrap sample (optimism = 0.011). The Control Model’s AUC was 0.802 (0.801 to 0.802) on the original sample and 0.811 (0.810 to 0.812) on the bootstrap sample (optimism = 0.009) (Appendix A).

## 4. Discussion

Our research question was whether the nodule types present in a CT scan had predictive value for the development of LC in another location. To investigate this, we performed weighted time-dependent regression analysis on a clinical case–cohort which only included patients with LC-free baseline CT scans. This was necessary to ensure the correct direction of causality: that the nodule types occurred before LC incidence. Age, sex, and other quantitative CT information were included in the analysis to control for potential confounding.

One of the cohort’s inclusion criteria was a high level of certainty that lung cancer was not (yet) detectable in the baseline scan. This can only be done retrospectively with information from follow-up scans; the models and predictors described should be validated and calibrated in a prospective clinical cohort.

Results of the univariable analysis and the Full Model revealed that the presence of at least one solid nodule, part-solid nodule, or ground-glass nodule were significant risk factors for future LC incidence (Appendix A, Figure 2). Parsimonious models were derived to simplify the interpretation by only considering the variables with the highest predictive power (Table 2 and Appendix A). The Nodule Type Model confirmed that the presence of nodules is a risk factor for future LC development. Though this observation has been previously described [4], a novel finding is that more part-solid and ground-glass nodules increase the risk linearly, whereas more solid nodules (beyond one) reduces the risk. As only 27 patients had more than two SSNs, it remains unclear whether the relationship is indeed linear for higher SSN counts. The presence of at least one perifissural nodule was a protective factor: its beta coefficient was equivalent in magnitude as that for the presence of at least one nodule (−1.11 and 1.12, respectively, Table 2), indicating that patients with nodules which are exclusively perifissural nodules have an equivalent risk to patients without any nodules.

By calculating the difference in AUC performance between the parsimonious models with and without nodule type information (Nodule Type Model and Control Model, respectively), we found that the added value of nodule-type information was 0.025. Note that the AUC is a summary measure that does not reflect clinical utility. To compensate, time-to-event curves were plotted to visually display the trend of higher LC risk among patients with more SSNs (Figure 3).

As would be expected from only including data from LC-free CT scans, the largest nodule diameter was not a predictor of future malignancy in multivariable analysis. In a scenario where LC may already be present, nodule size is otherwise established as the strongest predictor of LC based on a single scan [9,23,24]. No interaction was found between nodule type (i.e., solid vs. SSN) and nodule diameter.

The finding that the presence of an SSN may be a risk factor for future malignancy development in another lung location has thus far been described only once, by Silva et al. [4], based on results from the MILD screening population: of 389 screenees with at least one SSN, 30 LCs were diagnosed, of which 22 were not derived from an SSN. In this context, it is noteworthy that the same author group described an increased malignancy risk for patients with an overall increased lung density, interpreted as an increased inflammatory state [25].

We also found that greater numbers of solid nodules (beyond one) are indicative of lower LC risk. A possible explanation is that some patients naturally develop more solid nodules with no clinical significance. Most benign solid nodules are small (i.e., <6 mm) and represent benign entities [26]; not including small solid nodules in the count may counteract this finding. Note that there is a moderate inter-rater agreement when classifying solid nodules as perifissural nodules [27], and a lack of a standardized definition [14]. Therefore, there may have been an insufficient distinction between small solid nodules and perifissural nodules. We expect that a considerable number of these solid nodules could also be classified as perifissural nodules, of which the LC risk can be considered negligible [14].

The clinical implication of our results is that future LC risk models should consider incorporating SSN count. Such models would likely be most useful in a LC screening setting for the purpose of personalizing scan intervals. As our results were derived from a clinical cohort, subsequent research should first attempt to replicate our findings in a high-risk screening population. Although the predictive value of nodule-type information alone appears to be relatively small, it can be significant when applied to large numbers. To the best of our knowledge, no lung cancer risk model has yet considered the number of SSNs and solid nodules as variables [28]. Schreuder et al. [23] included the “presence of non-solid nodule” and “presence of part-solid nodule” in their parsimonious model for determining whole scan-based risk, but did not count the SSNs.

Another finding was that radiologist-rated presence of emphysema, bronchitis, interstitial lung disease, lymphadenopathy, calcifications of the coronary arteries, and calcifications of the transthoracic aorta are non-nodule CT features which contributed to LC prediction accuracy. This supports the notion that the presence of chronic obstructive pulmonary disease and/or cardiovascular disease may indicate susceptibility for LC, likely due to exposure to overlapping risk factors [24,29,30,31,32,33]. Note that while the presence of emphysema, coronary calcifications, and aorta calcifications was originally classified as “none”, “mild”, “moderate”, or “severe”, no statistically significant trends were found with increasing severity (results not shown); this was why these variables were converted into binary form for the analysis. While the relationship between these variables with LC incidence is unlikely to be linear [24], it is unclear whether quantitative CT measures would have been better predictors compared to clinician-reported outcome measures.

### Limitations

Our study has other limitations that should be taken into consideration. First, no data on smoking status or intensity were available, which would have been a strong demographic predictor of LC and may have been informative as a control variable or to select a higher risk population equivalent to that in a screening setting [34]. Data on patient mortality were also lacking, preventing the investigation of potential differences in LC prognosis; a higher LC risk does not imply a poorer prognosis [35]. Only the size of the largest nodule was considered for the model to avoid overfitting and to simplify model interpretation. The risk models only considered the earliest scan, from which it is not possible to derive information on nodule growth and whether nodules are transient or persistent. Finally, it is possible that the follow-up time was not sufficiently long, and that some LC nodules were erroneously deemed to be benign in our study.

## 5. Conclusions

In conclusion, our results indicate that the presence of single time-point SSNs on CT in a clinical setting may be a risk factor for future LC development in another lung location in a dose-dependent manner. The risk may increase linearly with the number of SSNs. This finding highlights the potential importance of utilizing information from the entire CT scan to improve the accuracy of LC risk assessment, and work towards personalized medicine.

## Figures and Tables

**Figure 1 cancers-13-02812-f001:**
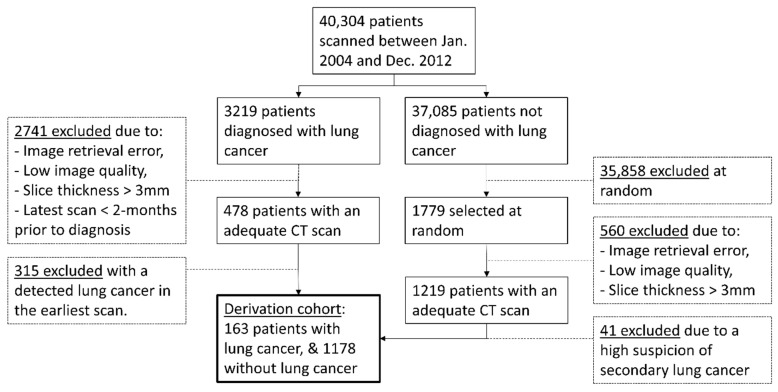
Patient selection flowchart.

**Figure 2 cancers-13-02812-f002:**
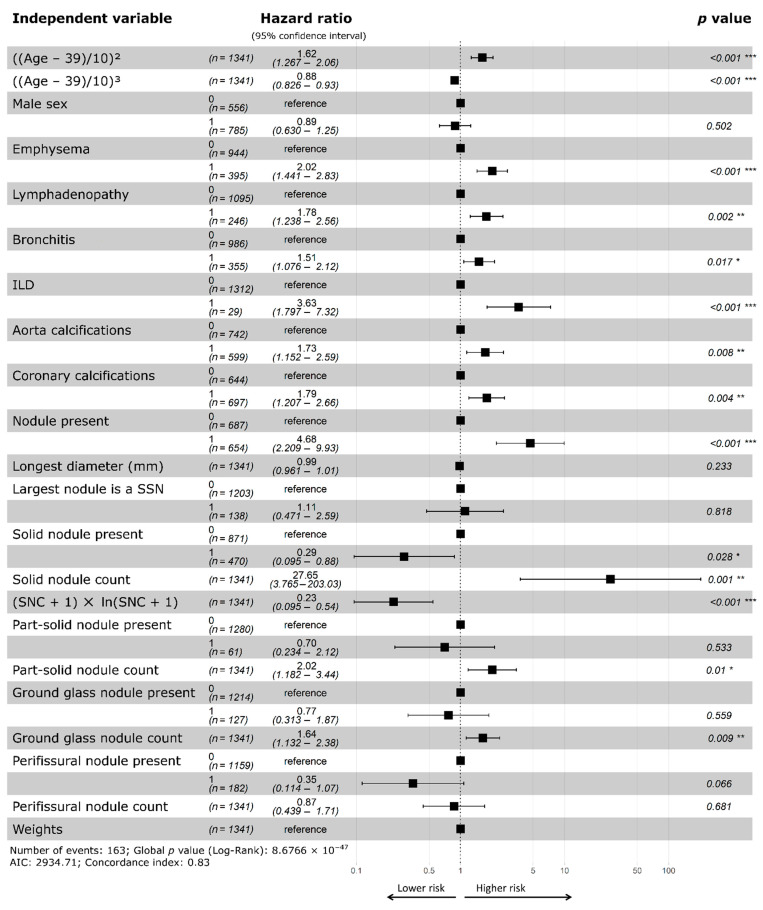
Forest plot of multivariable Cox regression. For binary variables, 0 (reference value) indicates absence and 1 indicates presence. For continuous variables (i.e., age, diameter, nodule counts), the hazard ratio refers to each unit increase. A hazard ratio greater than 1 indicates that the presence of the variable increases the risk of lung cancer incidence; conversely, a value less than 1 indicates that the presence of the variable reduces the lung cancer incidence risk. * *p* value < 0.05; ** *p* value < 0.01; *** *p* value < 0.001; ILD, interstitial lung disease; N, number of; PFN, perifissural nodule; SNC, solid nodule count; SSN, subsolid nodule.

**Figure 3 cancers-13-02812-f003:**
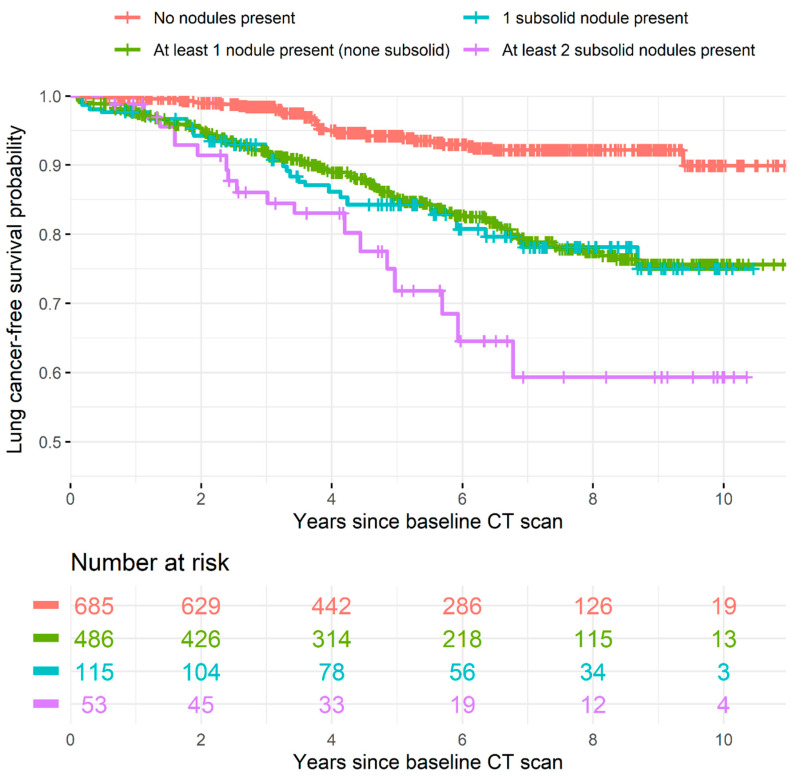
Lung cancer probability by nodule types present at baseline. This figure shows a Kaplan–Meier curve of lung-cancer-free survival, including a “number at risk” table. Vertical dashes on a line indicate the end of patients’ follow-up. The 5-year lung-cancer-free weighted survival probabilities were 0.942 (95% confidence intervals = 0.916, 0.969) when no nodules were present at baseline, 0.852 (0.817 to 0.887) when only solid nodules were present, 0.843 (0.775 to 0.917) when one SSN was present, and 0.718 (0.594 to 0.868) when at least two SSNs were present. The same figure with confidence intervals is provided as Appendix A.

**Table 1 cancers-13-02812-t001:** Cohort demographics.

Variable	Type	Categories	Lung Cancer Patients (*n* = 163)	Lung-Cancer-Free Patients (*n* = 1178)
Control patient characteristics				
Age at earliest scan, years	Continuous	N/A	64 (58–69)	61 (52–70)
Sex (male)	Binary	1 = Male; 0 = Female	100 (61)	685 (58)
Control CT features				
Emphysema	Binary	1 = Mild, moderate, or severe; 0 = None	88 (54)	307 (26)
Bronchitis	Binary	1 = Yes; 0 = No	68 (42)	287 (24)
Interstitial lung disease	Binary	1 = Yes; 0 = No	10 (6)	19 (2)
Lymphadenopathy	Binary	1 = Yes; 0 = No	48 (29)	198 (17)
Aortic calcifications	Binary	1 = Mild, moderate, or severe; 0 = None	107 (66)	492 (42)
Coronary calcifications	Binary	1 = Mild, moderate, or severe; 0 = None	115 (71)	582 (49)
Nodule present	Binary	1 = Yes; 0 = No	132 (81)	522 (44)
Mean diameter of largest nodule, mm	Continuous	N/A	9.0 (6.3–14.6)	6.8 (5.0–12.4)
Nodule type variables				
Largest nodule is a SSN	Binary	1 = Yes; 0 = No	36/132 (27)	102/522 (20)
Solid nodule present	Binary	1 = Yes; 0 = No	105 (64)	365 (31)
Solid nodule count	Continuous	N/A	1 (1–2)	1 (1–3)
Part-solid nodule present	Binary	1 = Yes; 0 = No	18 (11)	43 (4)
Part-solid nodule count	Continuous	N/A	1 (1–1)	1 (1–1)
Ground-glass nodule present	Binary	1 = Yes; 0 = No	31 (19)	96 (8)
Ground-glass nodule count	Continuous	N/A	1 (1–1)	1 (1–2)
Perifissural nodule present	Binary	1 = Yes; 0 = No	16 (10)	166 (14)
Perifissural nodule count	Continuous	N/A	1 (1–1)	1 (1–1)

Values given in brackets are percentages for binary variables and twenty-fifth and seventy-fifth percentiles for continuous variables. For continuous variables, only values greater than zero were summarized. N/A: not applicable.

**Table 2 cancers-13-02812-t002:** Parsimonious lung cancer risk model (Nodule Type Model).

Variable	Beta Coefficient	Hazard Ratio (95% Confidence Interval)	*p* Value
Patient characteristics			
Age at earliest scan, per year (((*x*−39)/10)^2^)	0.50255	1.65 (1.25 to 2.18)	<0.001
Age at earliest scan, per year (((*x*−39)/10)^3^)	−0.13941	0.87 (0.82 to 0.93)	<0.001
CT features excluding nodule type			
Emphysema	0.70903	2.03 (1.33 to 3.12)	0.001
Bronchitis	0.39078	1.48 (0.95 to 2.29)	0.08
Interstitial lung disease	1.24998	3.49 (1.36 to 8.97)	0.009
Lymphadenopathy	0.54126	1.72 (1.06 to 2.93)	0.028
Aortic calcifications	0.57696	1.78 (1.08 to 2.93)	0.023
Coronary calcifications	0.51209	1.67 (1.04 to 2.69)	0.035
Nodule present	1.12291	3.07 (1.54 to 6.13)	0.001
Nodule type information			
Solid nodule count, per nodule	1.39874	4.05 (1.30 to 12.63)	0.016
Solid nodule count, per nodule ((*x* + 1) × ln (*x* + 1))	−0.68170	0.51 (0.31 to 0.83)	0.007
Part-solid nodule count, per nodule	0.50634	1.66 (1.11 to 2.47)	0.013
Ground-glass nodule count, per nodule	0.41971	1.52 (1.06 to 2.17)	0.021
Perifissural nodule present	−1.11383	0.33 (0.16 to 0.66)	0.002

A parsimonious model was derived using Cox regression containing only variables with a *p* value < 0.20 using backwards elimination. Some variables were transformed as indicated by the mathematical expression in brackets, in which *x* represents the untransformed variable. N/A: not applicable.

## Data Availability

Customized data were supplied by the Netherlands Cancer Registry. Instructions for applying for data can be found here: https://iknl.nl/en/ncr/apply-for-data. It is also possible to consult the publicly accessible data available on www.kanker.nl (only in Dutch).

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
