# Peer review of "CT-Detected Subsolid Nodules: A Predictor of Lung Cancer Development at Another Location?"

_cancers, 2021, doi:10.3390/cancers13112812_

Round 1

Reviewer 1 Report

The manuscript is well-written, and the question being addressed is important. My main concern is that it is unclear if the lung cancers detected later are the actual SSN. Was it just that the nodules already represented atypical adenomatous hyperplasia or indolent lung cancers? If so, i'm not sure these are actually "predictive" of future lung cancer rather than suggestive of presence of multifocal lung cancer. Can the authors clarify if the radiologists reviewed/interpreted images independently, simultaneously, in tandem, or in consensus after discussion? In the methods it was noted that “nodules related to an obvious infection were excluded.” How was this determined? As there can be significant overlap in the imaging features of infection and malignancy in the absence of comparison studies, were any specific features used to determine this?

Author Response

Response to Reviewer 1 (cancers-1225266)

We thank the reviewer for their critique to help us to improve our manuscript.

The manuscript is well-written, and the question being addressed is important. My main concern is that it is unclear if the lung cancers detected later are the actual SSN. Was it just that the nodules already represented atypical adenomatous hyperplasia or indolent lung cancers? If so, I’m not sure these are actually "predictive" of future lung cancer rather than suggestive of presence of multifocal lung cancer.

We can indeed not guarantee that a subset of SSNs actually represents indolent cancers, however it is very unlikely that all SSNs seen in the study group were caused by solitary or multiple adenocarcinomas. Even if a subset of SSN will develop over a longer follow up time (our follow up time was at least 2 years) into an invasive adenocarcinoma, it does not change the message that the presence of one or multiple SSNs is associated with a greater lung cancer risk.

Can the authors clarify if the radiologists reviewed/interpreted images independently, simultaneously, in tandem, or in consensus after discussion?

We have clarified that the radiologists reviewed the images independently.

In the methods it was noted that “nodules related to an obvious infection were excluded.” How was this determined? As there can be significant overlap in the imaging features of infection and malignancy in the absence of comparison studies, were any specific features used to determine this?

We agree that there is a morphological overlap between persistent SSNs and temporary (infectious) SSNs (we have now included the following citation: Chung K, Ciompi F, Scholten ET, et al. (2018) Visual discrimination of screen-detected persistent from transient subsolid nodules: An observer study. PLoS ONE 13(2): e0191874. https://doi.org/10.1371/journal.pone.0191874). As obvious infections were classified as multifocal, unsharply defined, or confluent SSNs, typically seen in bronchopneumonia. Follow up scans were also used to differentiate persistent from temporary lesions. Nevertheless, we acknowledge that a subset of the SSNs seen in our study group may have been infections.

Reviewer 2 Report

The authors propose to evaluate whether the risk of having a nodule on a screening LC CT is a risk for developing another cancer.

The paper is well written and the analyses are  detailed and explained adequately. 

  1. The paper evaluates a cohort with history of lung cancer either current and or previous resection , hence the  conclusion is applicable to patients with prior history of lung cancer. It is a well known fact that people who get one cancer are at a greater risk of getting a second primary. Please modify title and your conclusions adequately.
  2. Can you clarify for the patients who were cancer free , what type of treatment they received for they cancer surgical or medical ?
  3. For the cohort with LC how was diagnosis made and how were they treated.
  4. Survival in LC is determined by the stage at presentation , hence  the survival for early cancer would be very different from stage 3.  Did you only include early stage  tumors if so  please add that to the inclusion criteria.
  5. The paper compares solid vs part solid nodules, and concludes that risk increases in a dose dependant manner , not clear what dose are they referring to?
  6. Even if smoking extent is not known, it would be important to know how many were current smokers and run a subgroup analyses for that cohort. 
  7. Solid perifissural nodules  are considered benign,  and it is unclear if they were included or not.
  8. They found that risk increases in  a linear fashion with increasing number of nodules. Hence the whole scan needs to be evaluated. The standard of care depend on evaluation of the whole scan and not on 1 image or nodule, and patients can have either 1 cancer or more than 1 primary cancer or mets or multifocal cancer , the prognosis is very different in all these scenarios and hence the distinction needs to be made by longitudinal follow up.
  9. One stratetgy would be to include only malignant nodules or nodules with malignant behaviour and exclude part solid or solid nodules that were benign 

Author Response

Response to Reviewer 2 (cancers-1225266)

We thank the reviewer for their critique to help us to improve our manuscript.

The authors propose to evaluate whether the risk of having a nodule on a screening LC CT is a risk for developing another cancer.

The paper is well written and the analyses are  detailed and explained adequately. 

1. The paper evaluates a cohort with history of lung cancer either current and or previous resection , hence the conclusion is applicable to patients with prior history of lung cancer. It is a well known fact that people who get one cancer are at a greater risk of getting a second primary. Please modify title and your conclusions adequately.

Perhaps we misunderstand this comment, but we did not focus on patients with a prior history of lung cancer. The scans were obtained retrospectively from a time point before the reported lung cancer diagnosis. We admit that it is possible that some patients may have been diagnosed with a malignancy at an earlier timepoint, but this information was not considered. As we would expect this to constitute a very small proportion of the patients, we do not believe that it is justified to explicitly mention this group of patients in our title and conclusions.

2. Can you clarify for the patients who were cancer free , what type of treatment they received for they cancer surgical or medical ?

As mentioned in the response to comment #1, this was not applicable.

3. For the cohort with LC how was diagnosis made and how were they treated.

As mentioned in the response to comment #1, this was not applicable.

4. Survival in LC is determined by the stage at presentation , hence  the survival for early cancer would be very different from stage 3.  Did you only include early stage  tumors if so  please add that to the inclusion criteria.

Survival was irrelevant in our study as the endpoint used was lung cancer diagnosis.

5. The paper compares solid vs part solid nodules, and concludes that risk increases in a dose dependant manner , not clear what dose are they referring to?

By “dose-dependent” we refer to the number of nodules, meaning that more nodules is associated with a higher risk of lung cancer.

6. Even if smoking extent is not known, it would be important to know how many were current smokers and run a subgroup analyses for that cohort.

Unfortunately, we also did not have information on which patients were current smokers.

7. Solid perifissural nodules  are considered benign,  and it is unclear if they were included or not.

One criteria for classifying a nodule as a perifissural nodule is that they are solid. So yes, solid PFNs were included in our study.

8. They found that risk increases in  a linear fashion with increasing number of nodules. Hence the whole scan needs to be evaluated. The standard of care depend on evaluation of the whole scan and not on 1 image or nodule, and patients can have either 1 cancer or more than 1 primary cancer or mets or multifocal cancer , the prognosis is very different in all these scenarios and hence the distinction needs to be made by longitudinal follow up.

We agree with this assessment, but this is outside the scope of our study because the assessment for lung cancer treatment/management would only occur after diagnosis. In a cancer-free patient, it is currently customary to assess each nodule individually for malignancy risk.

9. One stratetgy would be to include only malignant nodules or nodules with malignant behaviour and exclude part solid or solid nodules that were benign 

The aim of our study was to determine the risk of future lung cancer development based on nodule characteristics throughout the scan; excluding nodules would not conform with the purpose of our study. Furthermore, we made these predictions based on a cohort of scans in which we were very confident that no malignant nodules were present at the time.

Round 2

Reviewer 2 Report

The authors have addressed the concerns and satisfactorily revised the manuscript. 

Author Response

We thank the reviewer for their approval of our manuscript.